# Evaluating the Response of *Glycine soja* Accessions to Fungal Pathogen *Macrophomina phaseolina* during Seedling Growth

**DOI:** 10.3390/plants12223807

**Published:** 2023-11-09

**Authors:** Shirley Jacquet, Shuxian Li, Rouf Mian, My Abdelmajid Kassem, Layla Rashad, Sonia Viera, Francisco Reta, Juan Reta, Jiazheng Yuan

**Affiliations:** 1Department of Biological and Forensic Sciences, Fayetteville State University, Fayetteville, NC 28301, USA; sjacquet@broncos.uncfsu.edu (S.J.); mkassem@uncfsu.edu (M.A.K.); lrashad@broncos.uncfsu.edu (L.R.); viera_sonia@outlook.com (S.V.); francisocreta2@gamil.com (F.R.); jreta@broncos.uncfsu.edu (J.R.); 2Crop Genetics Research Unit, United States Department of Agriculture, Agricultural Research Service (USDA, ARS), 141 Experiment Station Road, P.O. Box 345, Stoneville, MS 38776, USA; shuxian.li@usda.gov; 3Soybean and Nitrogen Fixation Research Unit, United States Department of Agriculture, Agricultural Research Service (USDA, ARS), 3127 Ligon St., Raleigh, NC 27607, USA; rouf.mian@usda.gov

**Keywords:** *Macrophomina phaseolina*, *Glycine soja*, charcoal rot resistance, WinRHIZO

## Abstract

Charcoal rot caused by the fungal pathogen *Macrophomina phaseolina* (Tassi) Goid is one of various devastating soybean (*Glycine max* (L.) Merr.) diseases, which can severely reduce crop yield. The investigation into the genetic potential for charcoal rot resistance of wild soybean (*Glycine soja*) accessions will enrich our understanding of the impact of soybean domestication on disease resistance; moreover, the identified charcoal rot-resistant lines can be used to improve soybean resistance to charcoal rot. The objective of this study was to evaluate the resistance of wild soybean accessions to *M. phaseolina* at the seedling stage and thereby select the disease-resistant lines. The results show that the fungal pathogen infection reduced the growth of the root and hypocotyl in most *G. soja* accessions. The accession PI 507794 displayed the highest level of resistance response to *M. phaseolina* infection among the tested wild soybean accessions, while PI 487431 and PI 483660B were susceptible to charcoal rot in terms of the reduction in root and hypocotyl growth. The mean values of the root and hypocotyl parameters in PI 507794 were significantly higher (*p* < 0.05) than those of PI 487431 and PI 483460B. A analysis of the resistance of wild soybean accessions to *M. phaseolina* using the root and hypocotyl as the assessment parameters at the early seedling stage provides an alternative way to rapidly identify potential resistant genotypes and facilitate breeding for soybean resistance to charcoal rot.

## 1. Introduction

The cultivated soybean (*Glycine max* (L.) Merr.) is the domesticated variation of the wild soybean *Glycine soja* (Siebold & Zucc.). *G. soja* is found in broad ecological sites, from Far East Russia to South China [1]. However, *G. max* has lost much of its genetic diversity from the origin of *G. soja* through domestication and improvement processes [2]. It is generally believed that the domestication of *G. soja* occurred around 6000–9000 years ago in the areas along the Yellow River or Huang-Huai Valley in Central China [3]. It is also possible that the frequent introgressions between the wild and cultivated populations overlapped for a long period of time [4]. *G. max* is an economically important crop because of its high percentage of oil and protein in the seeds. Unlike *G soja*, the narrow genetic variation of *G. max* has limited the maximum yield along with the improvement in abiotic and biotic resistance [1]. Although both *G. max* and *G. soja* display apparent morphological distinction, both species share quite a similar genome structure and there is no reproductive isolation between them [5]. The extensive breeding effort selects high-yield variations and other desirable traits, including a proper stem architecture for machine harvesting as well as appropriate seed composition in *G. max*, which is not available in *G. soja*. With more than 22,000 accessions, including modern breeding lines and land race cultivars (*G. max*), *G. soja*, and perennial Glycine (www.soybase.org, 1 January 2021) have been collected and curated in the USDA Soybean Germplasm Collection [6], and a lot more accessions are stored in many private and public germplasm banks around the world. Therefore, *G. soja* possesses great potential for infinite gene pools of genetic diversity and environmental resilience to develop soybean varieties with anticipated traits, especially biotic resistance to soybean aphid, cyst nematode, and sudden death syndrome [1,7,8,9,10,11,12,13], and abiotic resilience, such as salt, temperature, and water stress tolerance [14,15,16,17,18]. These genetic resources are essential to optimize desired traits and improve soybean production. Soybean cyst nematode (SCN) is the most devastating pest in soybean production. However, the genetic variability in *G. max* for SCN resistance is limited, and the major allele for the resistance is derived from soybean germplasm PI 88788 [12]. To better decipher the genetic mechanism of SCN resistance in *G. max*, there is an urgent need to investigate the useful gene pool of *G. soja* to combat the devasting disease; the breeding effort has also been focusing on the unlimited genetic resources of *G. soja* accessions. A total of ten single-nucleotide polymorphism (SNP) markers were identified for SCN race 1 resistance in a genome-wide association study (GWAS) approach using the mapping panel of 1032 *G. soja* accessions, and three SNPs were in the same genomic position as previously defined QTLs [12]. More SCN resistance QTL were identified in *G. soja* accession PI 468916, and they were distinguished from that of QTLs found in *G. max*, which provided further evidence that wild soybean accessions possess the critical solution for the bottleneck issue of cultivated soybean [19]. The soybean aphid (*Aphis glycines*) is one of the devastating pests in soybean, which can cause significant yield loss if the pest has not been properly managed. The origin of *G. max* domestication and the infestation of soybean aphid have occurred in the overlapped regions implying that *G. soja* should possess the alleles of soybean aphid resistance. Two soybean aphid-resistant QTLs were identified in soybean chromosomes 8 and 16 in an advanced soybean breeding line, which derived from *G. soja* and explained 12.5–22.9% and 19.5–46.4% of phenotypic variations in trials over the years under greenhouse and field conditions, respectively [10]. However, the utilization of *G. soja* in soybean breeding programs is limited due to the genetic linkage of undesirable traits to beneficial alleles. With more than 1100 accessions in the USDA Soybean Germplasm Collection, *G. soja* still possesses a great potential for soybean selection and trait improvement, and the drawback of coinherited traits can also be prevented through several cycles of backcross with elite parental lines.

Charcoal rot (*Macrophomina phaseolina* (Tassi) Goid) is a soilborne disease of crops, which occurs in the mid to late season when plants generally suffer from excessive heat and drought stresses, and it has been widely spread to the soybean production area of the US states, such as Arkansas, Illinois, Indiana, Kansas, Kentucky, Missouri, Mississippi, and Tennessee [20,21,22,23,24,25]. The disease symptoms of charcoal rot are usually identified as light gray or silvery discoloration of the epidermal and subepidermal tissues on the lower stems and taproots of soybean plants in the field and generally emerge in R5-7 growth stages when soybean pots have undergone rapid seed filling [24]. The pathogen infection usually causes soybean plants to be wilted or stunted or to die prematurely, resulting in a significant yield loss [23,24,25,26,27,28]. A significant correlation has been described between drought stress and disease severity, which occurs in late development stages, and the severity of charcoal rot on soybeans worsens when the ground temperature is higher than 28 °C with limited soil moisture [24]. The pathogen of charcoal rot can survive on the plant debris as microsclerotia in the infected field for a long period of time even under inclement weather conditions. Charcoal rot infects the root tissue of a broad range of hosts, from corn to soybean plants within various growth stages and from seedlings to mature plants [28]. Currently, the commercially registered fungicides for *M. phaseolina* treatment are not available on the market [29]. The disease management strategies for soybean charcoal rot are not always successful due to the lack of elite-resistant cultivars on the seed market. The released cultivar DT 97-4290 has only provided moderate protection against charcoal rot; moreover, the soybean line PI 567562A showed better resistance to the pathogen than DT 97-4290 [30]. However, PI 567562A did not provide complete resistance to *M. phaseolina* because the pathogen still colonized roots but the colonization was less severe than other assessed genotypes in this study. Additionally, significant symptoms of bacterial blight, caused by *Pseudomonas savastanoi* pv. *glycinea* (Coerper), were observed and the disease severity was also impacted by soybean growth maturity, soil temperature, and air moisture during the development stages of soybean plants [30]. The efficacy of current strategies for charcoal rot disease management tactics is limited, and the cultural practices, such as a reduction in crop density, alternating planting dates, drought-tolerant cultivars, and irrigation are generally used to reduce disease severity [30]. Therefore, the unlimited genetic resources of *G. soja* will play an important role in addressing the availability of charcoal rot-resistant cultivars for host resistance. 

Root systems are the gateway to understand plant functions at both ecosystem to individual plant levels and thereby improve agricultural productivity and sustainability [31]. Studies have demonstrated that the introgression of the genes associated with root architecture Deeper Rooting 1 (*Dro1*) and phosphorous-starvation tolerance 1 (*Pstol1*) into rice can boost yield under environmental stresses, such as drought and low phosphorous, respectively [32,33], suggesting that these genes have a fundamental impact on plant development and production. The root, responsible for anchoring the plant to the soil, is an essential organ for overall plant growth and development. Roots offer crucial structural and functional support for the plants to acquire nutrients and water from the soil. Most root cells are derived from the meristematic cells of the root apex and then gradually differentiate into various types of cells in the spatial and functional distinct zones. The most straightforward method to assess root growth or development is to examine the length and areas of the primary root in seedlings. The best time to assess any root defects using the root parameters is about the 5th day after germination to determine damage from charcoal rot when root cells still display their embryonic origin [34]. Moreover, the abnormal growth of seedlings can be an indicator of the deficiency of overall plant growth and variability, which can impact plant growth and development within its life cycle. The assessment and characterization of different root parameters are therefore important for understanding plant growth [35]. As such, employing tools for precise, high-throughput phenotyping and the imaging of roots are very important in plant research and in the assessment of damage from disease. The current technology of image-based phenotyping has accurately assessed detailed structures of root systems in a relatively high-throughput manner. The technology can easily allow us to process many samples under both controlled and field environments. The root shape, size, and architecture of root systems are the primary targets for the assessment of the high-throughput, low-effort methods that obtain accurate information about root systems. As summarized by Topp et al. [31], numerous new root phenotyping systems have been developed. Some of the applications improve high throughput with the rich information content. The key achievements of these systems are the image-based quantification methods combining automation; they are thus less dependent on trained eyes, which is labor-intensive and cost-ineffective. The implementation of the imaging system that enables automatic and accurate analyses of many root samples with high throughput is crucial for reliable and efficient data collection and assessment. The software WinRHIZO Reg (www.regent.qc.ca) is designed to measure root phenotypes using several parameters. It is a reliable method to obtain the target traits in the roots of seedlings after the optimal parameters for root detection and identification have been programmed. The output of the assessed parameters of roots and hypocotyls has been simplified and minimized by the user intervention during the computation of target parameters within a low-scale and high-throughput manner. 

The objective of this study was to assay wild soybean accessions in response to *M. phaseolina* treatment at the seedling stage and thereby select the disease-resistant lines.

## 2. Materials and Methodology

### 2.1. Soybean Entries

All wild soybean accessions (*G. soja*) were obtained from the USDA Soybean Germplasm Collection (https://www.ars-grin.gov/npgs/) and cultivated and harvested at the Central Crop Research Station of North Carolina State University, Clayton, NC, USA. The Central Crops Research Station is located near the city of Clayton, North Carolina (35.66974° N and 78.4926° W). The average precipitation is 140.5 mm, and the mean daily temperature is about 24.5 °C from June to September. The elevation of this location is 81 m above sea level (Clayton, NC, USA, National Weather Service). Seven accessions of *G. soja* (PI 468916, PI 483460B, PI 487431, PI 507584, PI 507592, PI 507641, and PI 507794) were used to conduct the experiment. The seeds of *G. soja* accessions used for the experiment were from the 2021 harvest. 

### 2.2. Inoculum Preparation and Application

To prepare the inoculum of the fungal pathogen (*M. phaseolina* (Tassi) Goid), Difco Potato Dextrose Agar (PDA) was used to make the growth medium, which was adjusted to pH 4.8 with lactic acid. The Petri dishes and test tubes inoculated with the pathogen were placed in an incubator at 28 °C for approximately 2–3 weeks. The microsclerotia cultures were then washed with autoclaved distilled water, agitated to dislodge, and filtered with autoclaved four layers of cheesecloth into a sterilized 1000 mL glass bottle. The filtered microsclerotia suspension was diluted with the same volume of distilled water and stored in a refrigerator for future use. 

### 2.3. Seed Assays

Ten seeds from each *G. soja* (PI 468916, PI 483460B, PI 507593, PI 507584, PI 507592, PI 507641, and PI 507794) accessions were scarified with sandpaper. After the seeds were treated, they were soaked in small Petri dishes set with 10 mL of autoclaved distilled water and placed under aluminum foil for the 24 h germination process. Then, the seeds were transplanted on a 10 cM diameter Petri dish with filter paper and 6 mL of autoclaved distilled water as control. For inoculation treatment, seeds of *G. soja* were placed on the Petri dishes lined with filter paper and filled with 6 mL of the filtered *M. phaseolina* microsclerotia suspension prepared as described above. All Petri dishes were then sealed with parafilm and incubated in the dark for 72 h at room temperature. 

### 2.4. Data Collection and Analysis

Manual measurements of the hypocotyl and root were taken from all the seedlings using a ruler, and thereafter, each seedling was scanned and digital measurements were taken of the hypocotyl and root with the root parameters of Volume (cM^3^), Length (cM), Projected Area (cM^2^), Surface Area (cM^2^), Average Diameter (mm), and Length Per Volume (cM/m^3^) both from nontreated control and pathogen-treated samples using WinRHIZO Reg software (Regent Instruments, Quebec, QC, Canada) combining an Epson Expression 10,000XL scanner (Los Alamitos, CA, USA). The data were analyzed using various statistical tools, including boxplots, One-way Analysis of Variance (ANOVA), and Principal Component Analysis (PCA) using the R platform and visualized correlation matrix function [36]. The R package multcomp with Tukey function was used to compare the means of the assessed parameters among different lines including the pathogen treatment and untreated control.

## 3. Results and Discussions

### 3.1. Data Distribution and Comparison of Assessed Parameters

The plant root is one of the most important organs in plant growth and development due to its essential functionality and spatial anatomy [31]. It is critical to assess root growth under both pathogen-inoculated and noninoculated treatments to understand the impact of charcoal rot resistance at the early stage of plant development. Hence, there is a need for a tool to rapidly evaluate *G. soja* resistance to the disease. The seedlings of *G. soja* accessions are usually miniature and fragile; and therefore, it is very challenging to accurately measure the length of the root and hypocotyl manually with minimal user intervention. Five days after germination, the root length of *G. soja* was usually shorter than that of hypocotyl (Figure 1A). The manual measurements showed a similar growth rate among the *G. soja* control accessions ranging between 3 and 6 cM in the hypocotyl and 1 and 6 cM in the root (Figure 1). The distribution of the length of hypocotyl was normal; however, it appeared that the length of the root in the early growth stage was abnormally distributed. The various assessment parameters, including hypocotyl, root, and numerical data of Length (cM), ProjArea (cM^2^), SurfArea (cM^2^), AvgDiam (mm), LenPerVol (cM/m^3^), and RootVolume (cM^3^) respectively were collected from the disease-treated and nontreated control accessions using the software WinRHIZO combining an Epson Expression 10,000XL scanner. The boxplot displayed that the lengths of the root and hypocotyl were significantly (*p* < 0.05) decreased in the pathogen-treated accessions compared to that of the nontreated control (Figure 2A and 2B, respectively). The fungal pathogen infection significantly reduced most of the assessed parameters, except for the root width and surface area in the root (Figure 2A). It appeared that the total length of the root, measured both by using a ruler (C_Root and P_Root) and WinRHIZO (C_length and P-Length), root area (P_Area and P_Area), and length/volume had the highest impact in the assessment. A similar reduction was also observed in the hypocotyl; however, only the area and average diameter were increased by the pathogen treatment (Figure 2B) suggesting that the response to the pathogen infection in *G. soja* accessions appeared to display a high specificity for the assessed parameters in this experiment. Moreover, the pattern of reduction of each parameter was not precisely equal (Figure 2), suggesting that plant resistance to the pathogen was tissue-specific. As one of the worst fungal pathogens in soybean plants, the severe pathogenic impacts on the below-ground tissues of *G. soja* from *M. phaseolina* were observed at the early stage of plant growth, implying that disease treatment should be considered as early as possible. This in vitro experiment also provided a unique opportunity to investigate plant–pathogen interactions. The image-based system has the enormous potential to assess morphological variation with more accuracy and precision compared to the traditional measurement by using a ruler. However, the ability to acquire the correct data still needs a lot of effort in the student-centered project. Fortunately, the key feature of the WinRHIZO system is to combine the image-based quantification method and characterized algorithms together to illustrate the targeted root structure, which offers a convenient way to improve high-throughput and nondestructive measurement. Because the root system is subjected to continuous development in a temporal and spatial manner, uncovered specific morphological changes under disease pressure may provide crucial information to identify the association with the desired genotypes. 

### 3.2. Correlation among Assessed Parameters

The correlogram demonstrated a novel correlation among these assessed traits (Figure 3A,B). Based on the unassorted data (all lines were included), most of the accessed parameters were positively correlated with the other sister parameters (*p* < 0.001) among these accessions. However, the height and width, average diameter, and root volume were negatively correlated with each other, and the rest of the correlation coefficients showed predominantly positive values (Figure 3A). Moreover, the negative correlation was identified more frequently among the parameters in the hypocotyl than in that of the correlation in root parameters (Figure 3B). The ruler measured root length and showed a weak correlation between the root width and root area (r = 0.37 and r = 033, respectively) in contrast to the relationship between the hypocotyl area and width with a strong negative correlation (r = −0.96 and r = −0.99, respectively). The overall correlation pattern showed that most of the correlation coefficients of the root and hypocotyl parameters in the seedling stage were greater than 0.5 (*p* < 0.05). Note that this higher correlation in the assessed parameters did not necessarily reflect the organ-specific in the nontreated samples.

### 3.3. Multivariate Analysis

Further analysis was conducted using principal component analysis [37] to explore the multidimensional relationships between root and hypocotyl length with other root and hypocotyl parameters (Table 1 and Table 2). The PCA revealed the important correlations among the assessed root and hypocotyl parameters under laboratory conditions because PCA transferred a given set of the parameters that were correlated into principal components (PCs) and these PCs were not correlated and produced a series of eigenvalues and associated eigenvectors [37]. The more highly correlated the parameters, the fewer principal components were needed to sum up the initial data. The results show that the first two PCs in root parameters explained approximately 96% of phenotypic variance in the dataset. Except for the average diameter of the traits, most of the parameters contributed positively to the first principal component (PC1), which accounted for 91% of total phenotypic variance in root parameters. The rest of the PCs accounted for the remaining phenotypic variance, and the first four PCs accounted for 100% of phenotypic variance suggesting the power of PCA for the regrouping of existing variables (Table 1). The same result was obtained for the parameters of hypocotyl where the first PC explained 87% of phenotypic variation among the variables, and the second PC only explained 8% of phenotypic variation in the dataset. Most of the parameters in the first PC contributed to the PC positively, suggesting that these values should contribute to the PC same compared to the PC value. Interestingly, the results of this PCA indicate that most root and hypocotyl parameters had positive relationships with the PC1, which explained the major phenotypic variation, whereas they had a diverse relationship under PC2 and each variable contributed to the PC differently in PC2. A ggbiplot [38] was used to visually present the first two PCs (Figure 4A,B). In the PCA graphs, the length of the root and hypocotyl with width negatively contributed to the PC1 for indicating that the first principal component increased with the decreasing value of these two assessed parameters. All the assessed parameters that were grouped together are positively correlated to each other. Moreover, the parameters that were positively correlated should be displayed to the same sides of the biplot’s origin. The selected *G. soja* accessions with the resistant trait were clustered in the same quadrants (Figure 4A,B) in the analysis. This finding reflects that the tissues of root and hypocotyl display synchronized functionality in both control and pathogen treatments.

### 3.4. Selection of Resistant Line

In this study, seven accessions of *Glycine soja* (PI 468916, PI 483460B, PI 507593, PI 507584, PI 507592, PI 507641, and PI 507794) were used to evaluate the effects of the fungal pathogen *M. phaseolina* on the growth of root and hypocotyl in the seedling stage, which was an important and first step to assess *G. soja* plant resistance. The experiments showed a decrease in root and hypocotyl growth due to the pathogen inoculation in all accessions. PI 507794 appeared to have the highest level of resistance in response to *M. phaseolina* in terms of root and hypocotyl growth among the seven accessions, and PI 487431 and PI 483460B had the lowest levels of resistance to charcoal rot in the same tests (*p* < 0.05). The information on the charcoal rot resistance can be used to develop soybean line resistance to the fungal pathogen in breeding programs. Based on the data analyzed by WinRhizo software, the *G. soja* lines PI 483460B and PI 487431 were significantly susceptible to charcoal rot compared to the rest of the five *G. soja* accessions (Figure 5A,B). PI 468916, PI 507584, and PI 507641 showed moderate resistance to charcoal rot infection. Our results also suggest that sufficient phenotypical variations for charcoal rot resistance among these *G. soja* accessions can be identified as early as the seedling stage when reliable assessment methods are implemented under laboratory conditions. This artificial selection process allows us to detect the functional links between genotype and charcoal rot resistance. Combining ruler measurements and software quantification, the ability to link root parameters to their genetic attributes can be revealed. The advanced imaging tool of WinRHIZO system can be used to aid in selection of resistant soybean line. 

## 4. Conclusions

The analysis conducted in this study allowed us to identify potential charcoal rot-resistant lines in G. soja accessions by analyzing the assessed parameters of the root and hypocotyl at the early seedling stage. The mean values of the assessed parameters in PI 507794 were significantly higher (*p* < 0.05) than those of PI 487431 and PI 483460B. The assessed variables and their associated parameters can be used to select the elite lines with charcoal rot resistance and thereby provide useful information for disease-resistant trait improvement. The assessment method developed from this study provides an alternative way to rapidly identify potential resistant genotypes and facilitate breeding for resistance to charcoal rot.

## Figures and Tables

**Figure 1 plants-12-03807-f001:**
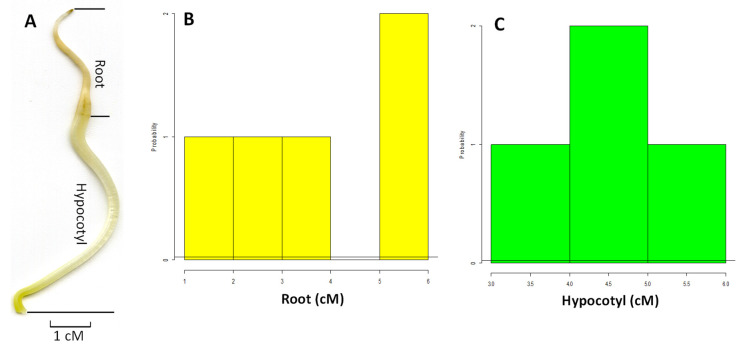
The wild soybean root and hypocotyl and distribution of hypocotyl and root as the assessed parameters measured by using a ruler. The unit of a ruler used in the assessment was centimeter (cM). (**A**). Anatomic sections of root and hypocotyl; (**B**). Distribution of the length of root; (**C**). Distribution of the length of hypocotyl.

**Figure 2 plants-12-03807-f002:**
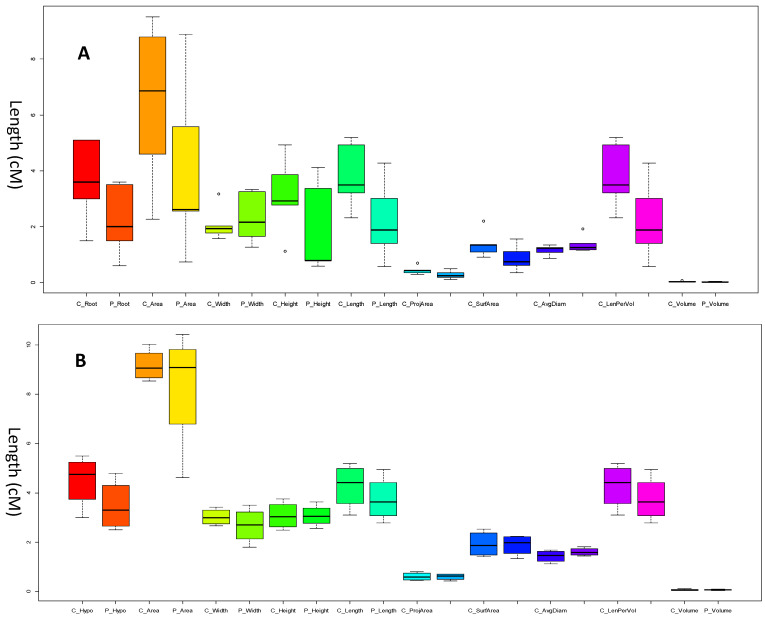
Comparison of assessed parameters of *G. soja* between nontreated control and charcoal rot treatment. The black bar in each boxplot presents the mean value of the assessed parameters. The abbreviation and units of assessment parameters were described as follows: C: control; P: pathogen treatment; Volume (cM^3^), Length (cM), ProjArea (cM^2^), SurfArea (cM^2^), AvgDiam (mm), and LenPerVol (cM/m^3^). (**A**). Room parameters; (**B**). Hypocotyl parameters.

**Figure 3 plants-12-03807-f003:**
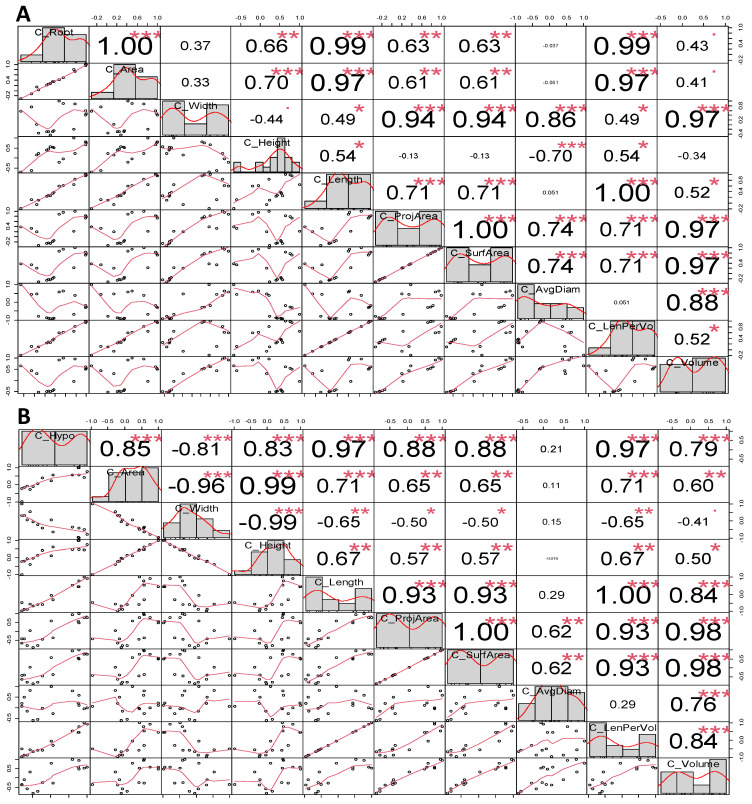
Correlations between assessed parameters under nontreated condition at room temperature. (**A**). Root parameters; (**B**). Hypocotyl parameters. Significance level: * *p* < 0.05, ** *p* < 0.01, and *** *p* < 0.001.

**Figure 4 plants-12-03807-f004:**
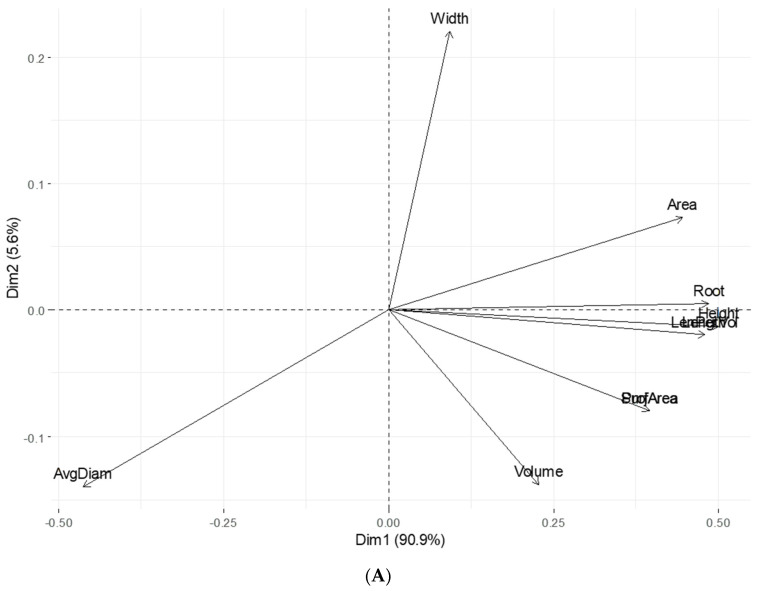
PCA of assessed parameters in *G. soja* line selection for charcoal rot resistance. The PCA graphs were made using ggbiplot of R program [30] based on prcomp algorithm (cran.r-project.org), and only PC1 vs. PC2 are presented. (**A**). PCA for root parameters; (**B**). PCA for hypocotyl parameters.

**Figure 5 plants-12-03807-f005:**
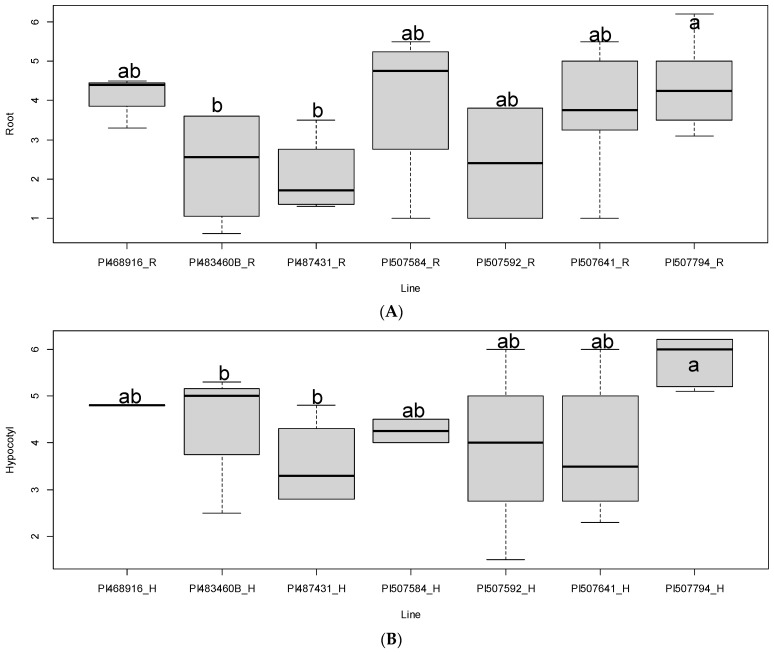
Comparison of *G. soja* lines for assessed parameters under charcoal treatment. (**A**). Root parameters; (**B**). Hypocotyl parameters. Different letters represented statistically different.

**Table 1 plants-12-03807-t001:** Contribution of the first five principal components to the root parameters using prcomp algorithm in R program and ggcorrplot package both the ggplot2 and visualized correlation matrix function (www.r-project.org).

	PC1	PC2	PC3	PC4
Root	0.37	0.01	0.20	0.73
Area	0.34	0.22	−0.06	−0.46
Width	0.07	0.68	−0.65	0.10
Height	0.38	−0.04	0.27	−0.28
Length	0.36	−0.06	−0.01	−0.19
ProjArea	0.30	−0.25	−0.24	0.02
SurfArea	0.30	−0.25	−0.24	0.02
AvgDiam	−0.35	−0.43	−0.34	−0.21
LenPerVol	0.36	−0.06	−0.01	−0.19
Volume	0.17	−0.42	−0.47	0.21
Eigenvalue	1.32	0.33	0.25	0.05
Proportion of Variance	0.91	0.06	0.03	0.00
Cumulative Proportion	0.91	0.96	1.00	1.00

**Table 2 plants-12-03807-t002:** Contribution of the first five principal components to the hypocotyl parameters using prcomp algorithm based visualized correlation matrix function in R program (www.r-project.org).

	PC1	PC2	PC3	PC4
Hypocotyl	0.40	0.07	0.05	0.35
Area	0.33	0.05	−0.28	−0.53
Width	0.19	−0.12	−0.84	−0.10
Height	0.35	0.17	0.41	−0.65
Length	0.38	−0.11	0.05	0.20
ProjArea	0.25	−0.36	0.09	0.06
SurfArea	0.25	−0.36	0.09	0.06
AvgDiam	−0.39	−0.60	0.03	−0.27
LenPerVol	0.38	−0.11	0.05	0.20
Volume	0.09	−0.56	0.13	−0.04
Eigenvalue	1.03	0.32	0.23	0.10
Proportion of Variance	0.87	0.08	0.04	0.01
Cumulative Proportion	0.87	0.95	0.99	1.00

## Data Availability

All authors agree with MDPI Research Data Policies.

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
