# Peer review of "Evaluating the Response of Glycine soja Accessions to Fungal Pathogen Macrophomina phaseolina during Seedling Growth"

_plants, 2023, doi:10.3390/plants12223807_

Round 1

Reviewer 1 Report

This research Assessed the Response to Charcoal rot  of Glycine soja at the Seedling Stage, and the results showed that the accession PI 507794 displayed the highest level of resistant response to M. phaseolina in terms of root and hypocotyl growth among tested wild soybean accessions, while PI 487431 was susceptible to charcoal rot in the same test. Analysis of the response of wild soybean accessions to M. phaseolina by using hypocotyl and root as the assessment parameters at the early seedling stage provides an alternative way to rapidly identify potential resistant genotypes and facilitate breeding for resistance to charcoal rot. IHere are some of my suggestions for this study.

1. The information in the title is incomplete. I suggest the author make modifications to better reflect the research results.

2. The content of the abstract is not specific, I suggest the author list the relevant data.

3. Please do not duplicate keywords with the title.

4.  In the introduction part, there are too much discussion of Soybean cyst nematode (SCN). I do not think it is necessary.

5. I suggest the author set up a control group (Glycine max)

6. Figure 1. B: Is the horizontal coordinate title "Hypocotyl (cM)"?

7. Did the author observe any symptoms of fungal infection during the experiment? 

8 Line 20: Please italicize "G. soja"

Author Response

  1. The information in the title is incomplete. I suggest the author make modifications to better reflect the research results.

We have revised the title:

Old title: Assessment of Glycine soja Accessions in Response to Charcoal rot (Macrophomina phaseolina) at the Seedling Stage

New Title: Evaluating the Response of Glycine soja Accessions to Fungal Pathogen Macrophomina phaseolina during Seedling Growth

  1. The content of the abstract is not specific, I suggest the author list the relevant data.

Has been modified based on the reviewer’s comment.

Old Abstract: Charcoal rot of soybean [Glycine max (L.) Merr.] caused by a fungal pathogen Mac-rophomina phaseolina (Tassi) Goid is an important disease which can severely reduce yield. Re-search of the wild soybean (Glycine soja) enhances the understanding of the course of the domes-tication history of the soybean and aids in development of resistant soybean genotypes to mitigate the pathogen dilemma.  The objective of this study was to assay wild soybean accessions in re-sponse to M. phaseolina at the seeding stage, and thereby select the disease resistant lines. The results showed that most of G. soja accessions had significant decrease in root and hypocotyl growth caused by charcoal rot.  The accession PI 507794 displayed the highest level of resistant response to M. phaseolina in terms of root and hypocotyl growth among tested wild soybean accessions, while PI 487431 was susceptible to charcoal rot in the same test. Analysis of the response of wild soybean accessions to M. phaseolina by using hypocotyl and root as the assessment parameters at the early seedling stage provides an alternative way to rapidly identify potential resistant genotypes and facilitate breeding for resistance to charcoal rot.

New Abstract: Abstract: Charcoal rot caused by the fungal pathogen Macrophomina phaseolina (Tassi) Goid is one of various devastating soybean [Glycine max (L.) Merr.] diseases, which can severely reduce crop yield. The investigation of genetic potential for charcoal rot resistance from wild soybean (Glycine soja) accessions will enrich our understanding of the impact of soybean domestication on disease resistance and moreover, the identified charcoal rot resistant lines can be used to improve soybean resistance to charcoal rot.  The objective of this study was to evaluate wild soybean accessions against M. phaseolina at the seedling stage, and thereby select the disease resistant lines. The results showed that the fungal pathogen infection reduced the growth of root and hypocotyl in most G. soja accessions. The accession PI 507794 displayed the highest level of resistant response to M. phaseolina infection among tested wild soybean accessions, while PI 487431 and PI 483660B were susceptible to charcoal rot in terms of the reduction of root and hypocotyl growth. The mean values of root and hypocotyl parameters, respectively in PI 507794 were significantly higher (P<0.05) than that of PI 487431 and PI 483460B. Analysis of the resistance of wild soybean accessions to M. phaseolina using root and hypocotyl as the assessment parameters at the early seedling stage provides an alternative way to rapidly identify potential resistant genotypes and facilitate breeding for soybean resistance to charcoal rot.

  1. Please do not duplicate keywords with the title.

Updated

  1. In the introduction part, there are too much discussion of Soybean cyst nematode (SCN). I do not think it is necessary.

Soybean cyst nematode (SCN) is considered as the top one soybean disease. The genetic resource from G. soja accessions is very important to combat SCN. Therefore, the description here for SCN resistance is the important example to address the significant relevant facts that the resistance of charcoal rot should be following.

  1. I suggest the author set up a control group (Glycine max)

The assessment for G. max for charcoal rot at seedling stage was conducted in the similar experiment but the morphological differences of root structures between two species were significant.  Therefore, we decided to present them in a separate manuscript in the future.

  1. Figure 1. B: Is the horizontal coordinate title "Hypocotyl (cM)"?

Changed

  1. Did the author observe any symptoms of fungal infection during the experiment?

Yes, the hyphae were often full of plates, but we discarded the plates if the significant amount hyphae in the plates were identified.

Reviewer 2 Report

The manuscript presents an important study on evaluating the response of wild soybean accessions to charcoal rot disease. However, there are several areas where the manuscript can be improved for suitability for its publication. Below are some critical points:

The title should be more concise and descriptive.

The abstract is informative but should be more concise and structured. It should include key results and conclusions.

Although the introduction provides background information on the importance of wild soybean (Glycine soja), it lacks a clear and focused research hypothesis or objective. A well-defined research question or objective should be explicitly stated.

The methods section should be more detailed and well-structured. It should provide step-by-step instructions on how the experiments were conducted, including the setup, conditions, and equipment used. Additionally, specific details about the statistical analyses performed should be included.

The discussion section should interpret the results and relate them back to the research question or objective. It should also discuss the significance of the findings in the context of existing literature. Furthermore, there is a lack of critical analysis of the limitations and potential biases of the study.

The manuscript should include more references or citations to support the statements made in the introduction and discussion. Properly citing relevant research is essential to provide context and credibility to the study. Followings are some of the important research papers which could be cited for this purpose

Javaid, A.; Khan, I.H. Chemical profile and antifungal activity of leaf extract of Tabernaemontana divaricata against Macrophomina phaseolina. Plant Prot. 2022, 6(3), 201–208.

Zafar, A.; Javaid, A.; Khan, I.H.; Ahmed, E; Shehzad, H.; Anwar, A. Synthesis of 4-hydroxyazobenzene, a promising azo dye for antifungal activity against Macrophomina phaseolina. Plant Prot. 2022, 6(2), 143–149.

The conclusion should be a concise summary of the key findings and their implications. It should also address the research objective stated in the introduction.

The references are overall correct and follow the format of the journal, however these should be crosschecked.

In summary, whereas the study addresses an important research question, the manuscript requires significant revisions to improve its organization, presentation of data, critical analysis, and overall language quality. Careful attention to these aspects will enhance the suitability of the manuscript for publication.

The manuscript needs significant improvement in terms of English language usage, grammar, and sentence structure. It is crucial to ensure that the language is clear and accurate for readers to understand the content easily.

Author Response

The abstract is informative but should be more concise and structured. It should include key results and conclusions.

We have revised it based on the reviewer’s comments.

Old Abstract: Charcoal rot of soybean [Glycine max (L.) Merr.] caused by a fungal pathogen Mac-rophomina phaseolina (Tassi) Goid is an important disease which can severely reduce yield. Re-search of the wild soybean (Glycine soja) enhances the understanding of the course of the domes-tication history of the soybean and aids in development of resistant soybean genotypes to mitigate the pathogen dilemma.  The objective of this study was to assay wild soybean accessions in re-sponse to M. phaseolina at the seeding stage, and thereby select the disease resistant lines. The results showed that most of G. soja accessions had significant decrease in root and hypocotyl growth caused by charcoal rot.  The accession PI 507794 displayed the highest level of resistant response to M. phaseolina in terms of root and hypocotyl growth among tested wild soybean accessions, while PI 487431 was susceptible to charcoal rot in the same test. Analysis of the response of wild soybean accessions to M. phaseolina by using hypocotyl and root as the assessment parameters at the early seedling stage provides an alternative way to rapidly identify potential resistant genotypes and facilitate breeding for resistance to charcoal rot.

New Abstract: Charcoal rot caused by the fungal pathogen Macrophomina phaseolina (Tassi) Goid is one of various devastating soybean [Glycine max (L.) Merr.] diseases, which can severely reduce crop yield. The investigation of genetic potential for charcoal rot resistance from wild soybean (Glycine soja) accessions will enrich our understanding of the impact of soybean domestication on disease re-sistance and moreover, the identified charcoal rot resistant lines can be used to improve soybean resistance to charcoal rot.  The objective of this study was to evaluate wild soybean accessions against M. phaseolina at the seedling stage, and thereby select the disease resistant lines. The results showed that the fungal pathogen infection reduced the growth of root and hypocotyl in most G. soja accessions. The accession PI 507794 displayed the highest level of resistant response to M. phaseolina infection among tested wild soybean accessions, while PI 487431 and PI 483660B were susceptible to charcoal rot in terms of the reduction of root and hypocotyl growth. The mean values of root and hypocotyl parameters, respectively in PI 507794 were significantly higher (P<0.05) than that of PI 487431 and PI 483460B. Analysis of the resistance of wild soybean accessions to M. phaseolina using root and hypocotyl as the assessment parameters at the early seedling stage provides an alternative way to rapidly identify potential resistant genotypes and facilitate breeding for soybean resistance to charcoal rot.

Although the introduction provides background information on the importance of wild soybean (Glycine soja), it lacks a clear and focused research hypothesis or objective. A well-defined research question or objective should be explicitly stated.

We have made modifications.

The methods section should be more detailed and well-structured. It should provide step-by-step instructions on how the experiments were conducted, including the setup, conditions, and equipment used. Additionally, specific details about the statistical analyses performed should be included.

We have added related information based on the reviewer’s suggestions.

The discussion section should interpret the results and relate them back to the research question or objective. It should also discuss the significance of the findings in the context of existing literature. Furthermore, there is a lack of critical analysis of the limitations and potential biases of the study.

It has been improved.

The manuscript should include more references or citations to support the statements made in the introduction and discussion. Properly citing relevant research is essential to provide context and credibility to the study. Followings are some of the important research papers which could be cited for this purpose

We have added the contents based on the reviewer’s suggestions.

Javaid, A.; Khan, I.H. Chemical profile and antifungal activity of leaf extract of Tabernaemontana divaricata against Macrophomina phaseolina. Plant Prot. 2022, 6(3), 201–208.

 Zafar, A.; Javaid, A.; Khan, I.H.; Ahmed, E; Shehzad, H.; Anwar, A. Synthesis of 4-hydroxyazobenzene, a promising azo dye for antifungal activity against Macrophomina phaseolina. Plant Prot. 2022, 6(2), 143–149.

The conclusion should be a concise summary of the key findings and their implications. It should also address the research objective stated in the introduction.

We have made modifications.

The references are overall correct and follow the format of the journal, however these should be crosschecked.

It has been checked.

In summary, whereas the study addresses an important research question, the manuscript requires significant revisions to improve its organization, presentation of data, critical analysis, and overall language quality. Careful attention to these aspects will enhance the suitability of the manuscript for publication.

Old conclusion: The analysis conducted in this study allowed us to analyze the relationship of assessed parameters for the root and hypocotyl at the early seeding stage. The two assessed variables and their associated parameters can be used to select the elite lines with target traits, and thereby provide useful information for disease resistant trait improvement. The assessment method developed from this study provides an alternative way to rapidly identify potential resistant genotypes and facilitate breeding for resistance to charcoal rot.

New conclusion: The analysis conducted in this study allowed us to identify charcoal rot-resistant lines in G. soja accessions by analyzing the assessed parameters of the root and hypocotyl at the early seedling stage. The mean values of assessed parameters, respectively in PI 507794 were significantly higher (P<0.05) than that of PI 487431 and PI 483460B. The assessed variables and their associated parameters can be used to select the elite lines with charcoal rot resistance, and thereby provide useful information for dis-ease-resistant trait improvement. The assessment method developed from this study provides an alternative way to rapidly identify potential resistant genotypes and facilitate breeding for resistance to charcoal rot.

Comments on the Quality of English Language

The manuscript needs significant improvement in terms of English language usage, grammar, and sentence structure. It is crucial to ensure that the language is clear and accurate for readers to understand the content easily.

Indeed, we didn’t pay much attention to it. The proofreading has been conducted.

Round 2

Reviewer 1 Report

I think this manuscript has been well revised. The author answered all my questions. I don't have any more comments.

The English expression of this manuscript is good. I have no suggestion for revision.